# The Effect of Molecular Weight on the Antibacterial Activity of *N*,*N*,*N*-Trimethyl Chitosan (TMC)

**DOI:** 10.3390/ijms20071743

**Published:** 2019-04-09

**Authors:** Priyanka Sahariah, Dorota Cibor, Dorota Zielińska, Martha Á. Hjálmarsdóttir, Dawid Stawski, Már Másson

**Affiliations:** 1Faculty of Pharmaceutical Sciences, School of Health Sciences, University of Iceland, Hofsvallagata 53, IS-107 Reykjavík, Iceland; prs1@hi.is; 2Department of Material and Commodity Sciences and Textile Metrology, Lodz University of Technology, 90-924 Lodz, Poland; dorowoj@gmail.com (D.C.); d.zielinska85@gmail.com (D.Z.); dawid.stawski@p.lodz.pl (D.S.); 3Instiute of Security Technologies “MORATEX”, Laboratory of Chemistry, 90-505 Lodz, Poland; 4Faculty of Medicine, Department of Biomedical Science, University of Iceland, Stapi, Hringbraut 31, 101 Reykjavík, Iceland; hjalmars@hi.is

**Keywords:** acidic hydrolysis, *N*,*N*,*N*-trimethyl chitosan, chitosan, average molecular weight, antibacterial activity

## Abstract

*N*,*N*,*N*-trimethyl chitosan (TMC) with 93% degree of trimethylation was synthesized. TMC and the chitosan starting material were subjected to acidic hydrolysis to produce 49 different samples with a reduced average molecular weight (M_w_) ranging from 2 to 144 kDa. This was done to allow the investigation of the relationship between antibacterial activity and M_w_ over a wide M_w_ range. NMR investigation showed that hydrolysis did not affect the degree of trimethylation (DS_TRI_) or the structure of the polymer backbone. The activity of TMC against *Staphylococcus aureus* (*S. aureus*) increased sharply with M_w_ until a certain M_w_ value (critical M_w_ for high activity, CMW) was reached. After the CMW, the activity was not affected by a further increase in the M_w_. A similar pattern of activity was observed for chitosan. The CMW was determined to be 20 kDa for TMC and 50 kDa for chitosan.

## 1. Introduction

Chitosan is a biopolymer obtained from chitin. Chitin can be processed in different ways to produce chitosan having varying degrees of acetylation (DA) and chain length or weight average molecular weight (M_w_). Several studies reported so far produced varying M_w_ ranges for chitosan and chitooligosaccharides (COS, M_w_ < 20 kDa) [1,2,3]. Low M_w_ chitosan or COS possess a wide range of biological activities and potential applications in drug delivery [4], including antimicrobial, antioxidant, hemostatic, wound healing, hypoglycemic and enzyme inhibition activities, and various other applications [4].

A number of studies reported the effect of M_w_ on the antimicrobial activity of chitosan and COS. Kulikov et al. [5] investigated COS within a M_w_ range of 3.5–20 kDa for activity against methicillin-resistant *Staphylococcus aureus* (MRSA), within a pH range of 5.5–6.75. They have also explored the effect of M_w_ on the growth and viability of a series of *Candida* species using COS in the M_w_ range 5.9–600 kDa [6]. Chang et al. [7] studied chitosan and COS with M_w_ values of 3–300 kDa for antibacterial efficacy against *Staphylococcus aureus* (*S. aureus*) and *Escherichia coli* (*E. coli*), at pH 5 to 7, and at different temperatures (4–45 °C). The effect of size and pH on antibacterial activity of COS with M_w_ 0.34 to > 2.0 kDa (Dp 2 to > 12) was reported by Li et al. [8] No et al. [9] reported the variation in antibacterial activity of chitosan towards 11 strains of bacteria with M_w_ varied from 28–1671 kDa and Mellegård et al. [10] *Bacillus cereus*, four strains of *E. coli* and four strains of *S. typhmurium* for chitosan with M_w_ 2–224 kDa. Younes et al. [11] investigated chitosan with Mw from 42 to 135 kDa and different degrees of acetylation (DA 2–61%) against four Gram-negative bacteria and three fungi. Omura et al. [12] have also studied chitosan oligomers with M_w_ 0.18–0.99 kDa, as well as chitosan with different DA and M_w_ (6–400 kDa) for activity against 9 strains of bacteria and yeast. 

Notwithstanding this significant number of studies, there appears to be no clear consensus on the general relationship between M_w_ and antimicrobial activity of chitosan and COS. Thus these authors have concluded that COS below a certain molecular weight are inactive whereas COS with higher molecular weights are highly active [5]; that there is a positive relationship between Mw and activity at acidic pH but a negative relationship at neutral pH [7]; that 0.8 kDa M_w_ (Dp 5) is needed for activity and that inhibitory effect increases with M_w_ [8]; that 470 kDa chitosan is generally most active with few exceptions [9]; that M_w_ has no effect on activity [10]; that there is a positive relationship for Gram-positive bacteria and a negative relationship for Gram-negative bacteria [11]; or that the minimum MIC is at M_w_ 11–30 kDa [12]. 

The solubility and antimicrobial activity of chitosan can be significantly enhanced by chemical modification [13]. Numerous antimicrobial chitosan derivatives have been reported but in contrast to the substantial number of studies on unmodified chitosan, there are few studies that have investigated the effect of M_w_ of chitosan derivatives on antibacterial activity. Runarsson et al. [14] reported the effect of methylated COS (M_w_ = 8.1 kDa and 0.77 kDa; a degree of quaternization = 0–74%) on *S. aureus* at pH values of 5.5 and 7.2. The study showed that at pH = 5.5, low M_w_ chitosan and quaternized chitosan (M_w_ = 8.1 kDa) were highly active, while at pH = 7.2, only the highly *N*-quaternized chitosan was active. On the other hand, the low M_w_ COS (M_w_ = 0.77 kDa) remained inactive towards *S. aureus* at both pH values. In a further study *N*-(2-(*N*,*N*,*N*-trimethylammoniumyl)acetyl and *N*-(2-(*N*-pyridiniumyl)acetyl) derivatives of glucosamine, chitooligomer, and chitosan were investigated for activity against *S. aureus* (MRSA), *E. faecalis* and *P. aeruginosa* [15]. It was found that the glucosamine derivatives were inactive (MIC ≥ 8192 µg/mL), the COS derivatives had low to moderate activity (MIC = 4096 to 512 µg/mL) and the chitosan derivatives had moderate to high activity (MIC = 512 to 32 µg/mL). We have also previously published a study on fully quaternized *N*-acyl chitosan derivatives within the M_w_ range of 19.8–13.2 kDa and DA 7–34% [16]. This study showed that the activity towards *S. aureus* and *E. coli* was highly dependent on the chain length separating the cationic charge from the polymer. In contrast, the variation of antibacterial activity with M_w_ or DA was not considered significant in this study, as the MIC values generally varied only by 1–2 dilutions (i.e., 2–4 fold difference in MIC) for different M_w_ and DA. The effect of aminoethyl modified chitosan (DS = 1.7–1.9) with varying M_w_ (99, 51, 27 and 1.4 kDa) has also been investigated towards *E. coli* at different pH values [17]. The study concluded that within the M_w_ range 99–27 kDa, the derivatives displayed similar high activity (MIC = 4–16 µg/mL), while derivatives with M_w_ of 1.4 kDa had significantly lower antibacterial activity. 

These previous studies on the effect of M_w_ variations on the antimicrobial activity of chitosan derivatives are few in number and much more limited than similar studies for unmodified chitosan. Hence, the purpose of the current study was to investigate a wider M_w_ range for an antimicrobial chitosan derivative. TMC, with a high degree of quaternization and low O-methylation, was chosen for this study as previous investigations have shown this derivative is highly active, especially against the Gram-positive bacteria *S. aureus* [14,15,16,18,19].

The effect of M_w_ on the antibacterial activity of TMC was to be investigated and as a control, we would also compare the results with a similar M_w_ variation in chitosan. We utilized acidic hydrolysis to obtain a large number of samples with different M_w_ for each structure. The hydrolysis of TMC and chitosan was performed under mild and strong acidic conditions at an optimized temperature. We then studied the rate of M_w_ reduction under both conditions and identified the obtained hydrolyzed products by different spectroscopic techniques. The obtained wide range of M_w_ for TMC and chitosan was then utilized for investigating the relationship between antibacterial activity (MIC) toward *S. aureus* and M_w_. 

## 2. Results

### 2.1. Synthesis

Several studies reported the quaternization of chitosan at the 2-amino position and the use of various reaction conditions resulted in obtaining low to high degree of substitution for trimethylation (DS_TRI_). Increasing the reaction time or repeated addition of the reagents to increase the DS_TRI_ is generally accompanied by an increase in *O*-methylation in the products. In our previously published method [14,20], we have shown that the use of MeI in a DMF-H_2_O solvent system (Scheme 1) can be used to obtain high DS_TRI_, without any significant *O*-methylation. Hence, we utilized this method for the synthesis of the TMC used in this study. Chitosan was treated with the aforementioned reaction conditions before reprecipitation and repeated addition of the reagents. The spectra for the TMC synthesized can be seen in Figure 1A and the protons appear as follows: GlcNAcCH_3_ (2.01 ppm), GlcN(CH_3_)_3_ (3.28 ppm), GlcN+GlcNAc H-2, H-5, H-6 (3.68–4.07 ppm), GlcN+GlcNAc H-3, H-4 (4.30–4.42 ppm) and GlcN H-1 (5.42 ppm). Small peaks, showing low levels of 3-O and 6-O methylation, are observed at 3.38 and 3.46 ppm respectively. The DS_TRI_ was calculated from the integrals of the ^1^H-NMR spectrum (Figure 1) and was found to be 92.6%. The degree of substitution for dimethylation (DS_DI_) was 3% with traces of *O*-methylation (DS_O-methyl_ = 2%). 

In the spectrum for starting chitosan (Figure 1B), the protons appear as follows: GlcNAcCH_3_ (2.08 ppm), GlcN+GlcNAc H-2 (3.21ppm), GlcN+GlcNAc H-3-H-6 (3.78–3.93 ppm), GlcNAc H-1 (4.79 ppm) and GlcN H-1 (4.92 ppm, overlapped with the HOD peak).

### 2.2. Acidic Hydrolysis

Previous studies reported the hydrolysis of chitosan to produce low M_w_ chitosan and COS, mostly using chemical reagents, including treatment with acids and oxidative-reductive depolymerization. The use of hydrochloric acid for hydrolysis of chitin and chitosan has been reported in a few studies [1,3,6]. The hydrochloric acid catalyzed the transformation of chitin consists of two main reactions, hydrolysis of the glycosidic linkage and the *N*-acetyl linkage [4]. Hydrochloric acid has been used in different concentrations (0.5 M–12 M) and temperatures (40–100 °C) to produce chitosan and chitooligomers having different ranges of M_w_ 7–2 [21], 37–15 [21], 40–20 [22], 64–7.5 [23], etc. Several other acids were also reported for chitosan hydrolysis, but they usually suffer from certain drawbacks. Sulfuric acid produces drastic hydrolysis but with sulfation and nitrous acid causes hydrolysis without deacetylation but involves the formation of 2,5-anhydro-d-mannose at the new reducing end [4,24,25]. On the other hand, fluorohydrolysis of chitosan results in the introduction of a fluoro group at the anomeric carbon of chitosan or oligomers [4,26]. Acidic hydrolysis of chitosan by 85% phosphoric acid was found to be dependent on the temperature and time of the reaction [2] while degradation of chitosan using an aqueous solution of acetic acid requires high temperature and high acid concentrations [27]. Reviewing the results from these studies, we concluded that hydrolysis of chitosan using hydrochloric acid would be the best choice for hydrolyzing the glycosylic bond without causing other changes in the chemical structure. 

The acid concentration can be expected to have an effect on the hydrolysis of chitosan, as seen in previous studies [1]. Hence, we performed the acidic hydrolysis of chitosan and TMC using HCl at two different concentrations: 1 M (dilute) and 12 M (concentrated). The hydrolysis of TMC and chitosan using 1 M HCl resulted in polymers having a M_w_ range of 144–60 kDa (TMC) and 138–35 kDa (chitosan) when the hydrolysis time was set between 0.25–206 h (Table 1). While the use of 12 M HCl resulted in faster hydrolysis within the same time interval, the lowest M_w_ values obtained for TMC and chitosan and were 3 kDa and 2 kDa respectively. The percentage reduction in the M_w_ value is plotted against time in Figure 2. Treatment with 1 M HCl at 60 °C resulted in a significant reduction in the M_w_ of TMC, with M_w_ decreasing by 22% of the original value (186 kDa) within the first 0.25 h. After 0.25 h, the M_w_ decreased slowly and reached a maximum reduction of 67%. On the other hand, treatment with 12 M HCl resulted in a M_w_ reduction of 47% and 98% after similar time intervals. In the case of chitosan, the hydrolysis was seen to follow a similar pattern

Initially, TMC and chitosan were treated with 1 M and 12 M HCl at 30 °C. When the hydrolysis of chitosan was performed with 1 M HCl at a higher temperature of 60 °C, the rate was found to increase by 2–7 fold. However, a further increase in temperature to 80 °C did not result in a significant increase in the hydrolysis rate in the case of chitosan. Hence, for 1 M HCl, the optimized temperature for the hydrolysis of TMC and chitosan was 60 °C. In the case of 12 M HCl, increasing temperatures above 30 °C resulted in a color change of the materials; therefore, for 12 M HCl, the optimal temperature was 30 °C. 

### 2.3. Structural Characterization of Hydrolyzed Products

The depolymerized material was investigated by ^1^H-NMR, COSY-NMR and FT-IR spectroscopy to determine if any structural changes in the polymer chain occurred during the hydrolysis process. The ^1^H-NMR spectra for the anomeric region of the hydrolyzed TMCs are shown in Figure 3A, where the α-anomeric reducing end protons for the GlcN unit appear at 5.63 ppm. The intensity of this proton is seen to increase with the increase in the hydrolysis time. The ^1^H-NMR spectra for the anomeric region of the chitooligomers obtained after time intervals 1 h, 4h, and 48 h are shown in Figure 3A. The α-anomeric reducing end protons appear at 5.47 ppm for the GlcN unit and at 5.23 for the GlcNAc unit. With a decrease in the length of the polymer chain, the intensity of the GlcN H-1α proton increases and becomes dominant, while the GlcNAc H-1α proton remains constant in all the spectra and could be detected in only traces, due to low DA in the polymer. The COSY-NMR spectra for the TMC and chitosan oligomer (Figure 3C,D) further confirm that the structural units of the polymers remain unaltered during the hydrolysis process.

The structural characterization of the TMC and chitosan and their hydrolyzed products were also performed using FT-IR spectra. Figure 4A shows an overlay of TMC and chitosan spectra obtained for samples taken at different times of hydrolysis. The IR frequencies are as follows: 3239 (O–H), 2927 (C–H), 1636 (C=O amide I), 1473 (C–H), 1030 (C–O–C) and 880 (C–H for polysaccharide structure) cm^−1^. When compared to the chitosan (Figure 4B) spectrum, it is seen that the peak in the region 1508 cm^−1^ is absent. This is due to the absence of the N–H bending of the primary amine in TMC. However, no characteristic peak for the NMe_3_ group could be observed in the IR spectrum of TMC. No significant changes in the IR spectra for the TMC oligomers could be seen, confirming that TMC structure remains unchanged during the acidic hydrolysis process. In Figure 4B, the IR frequencies for chitosan are as follows: 3239 (O–H), 2879 (C–H stretching), 1617 (C=O amide I), 1508 (N–H bending), 1150 (C–O glycosidic linkage), 1062 (C–O–H, C–O–C and CH_2_CO) and 895 (C–H for polysaccharide structure) cm^−1^. The spectra for chitosan obtained after 1 h, 4 h, 48 h and 528 h show peaks at similar regions without the appearance of any new peaks, confirming that the structure remains intact during the hydrolysis process. The only difference in the spectra was that the peaks which were broadened in the case of chitosan were more sharpened in the case of the oligomers, due to the decrease in the size of the polymer chains. 

### 2.4. Mw-Antibacterial Property Relationship

The hydrolyzed TMC and chitosan obtained within the M_w_ range of 144–3 kDa and 138–2 kDa, respectively, and their starting materials (TMC = 186 kDa and chitosan = 225 kDa) were investigated for antibacterial activity towards *S. aureus*. The aim was to investigate whether the M_w_ change within this range would have a significant impact on the antibacterial activity of TMC and chitosan. There was a 1000-fold variation in the measured activity in this study (Table 2). This data was therefore plotted as log(1/MIC) (MIC (g/mL) on a reverse logarithmic scale) versus M_w_, as shown in Figure 5. The TMC and depolymerized TMC showed a clear correlation between Mw and activity against *S. aureus* (Figure 5A). Within a M_w_ range of 144–20 kDa, the materials showed high activity, with log(1/MIC) value of 4.5 (MIC= 32 × 10^−6^ g/mL = 32 µg/mL) in most cases. The antibacterial activity, however, decreased with decreasing M_w_ when the M_w_ value was less than 20 kDa and reached a minimum value of log(1/MIC) = 1.48 (MIC ≥ 32,768 µg/mL), at M_w_ = 3 kDa. 

As a control, a similar study was done with the depolymerized chitosan samples. Figure 5B shows chitosan is highly active towards *S. aureus* within the M_w_ range of 200–50 kDa. No significant difference in the activity of the polymers was observed for this M_w_ range and the log(1/MIC) value was 2.4 (MIC = 4096 µg/mL). With further lowering of the Mw value (50–10 kDa), the activity lowers and the log(1/MIC) value drops to 1.0 remaining constant within this range. Chitosan oligomers with Mw 2–4 kDa did not show any measurable activity and were therefore at least four times less active. To determine the antimicrobial efficacy of the lowest unit, glucosamine (M_w_ = 0.179 kDa) was also tested against *S. aureus*, but this monomer did not show any activity within the measured range of dilutions (MIC ≥ 32,678 µg/mL). 

Thus, in both cases, the activity of the oligomer/polymer increases with M_w_ until a certain value is reached, which we have defined as critical mole weight for high activity (CMW). After this, the activity is virtually independent of M_w_. According to Figure 3A the CMW for TMC is ~20 kDa and according to Figure 3B, the CMW for chitosan is ~50 kDa.

## 3. Discussion

### 3.1. Synthesis and Acid Degradation

The procedure for the *N*,*N*,*N*-trimethylation of chitosan has been previously reported [20,28,29]. By repeated methylation reactions a product with a very high degree of trimethylation (quaternization) and a low degree of O-methylation was obtained. 

The hydrolysis of TMC and chitosan was carried out in acidic medium. The temperature of the medium is often found to have a significant effect on the acidic hydrolysis of chitosan. In this study, it was found that the optimal temperature for the hydrolysis of chitosan and TMC was 60 °C. For 12 M HCl, the optimal temperature was 30 °C to avoid color change of the product. A previous study reported that the hydrolysis of chitosan oligomers using HCl showed a 6-fold increase from 3 M to 6 M acid concentration and another 6-fold increase from 6 M to 12 M acid concentration. Furthermore, when the reaction temperature was increased from 25 °C to 30 °C and from 30 °C to 35 °C, the rate of hydrolysis increased by 2-fold each time [1].

### 3.2. M_w_-Antibacterial Property Relationship

This is the first systematic study of the M_W_ activity relationship for antimicrobial chitosan derivative where there is both a high degree of substitution (93% *N*,*N*,*N*-trimethylation) and a wide range of M_w_ (186–3 kDa) in the investigated products. The current results are generally consistent with our previous studies of highly substituted N-acyl quaternary chitosan derivatives [16] where the activity was found to be independent of M_w_ in the M_w_ range 13–20 kDa, which is consistent with a slightly lower CMW for N-acyl derivatives than for TMC. The earlier study of similar *N*-(2-(*N*,*N*,*N*-trimethylammoniumyl)acetyl and *N*-(2-(*N*-pyridiniumyl)acetyl) derivatives found that glucose amine derivatives were inactive (MIC ≥ 8192 µg/mL) towards *S. aureus* (MRSA), *E. faecalis* and *P. aeruginosa,* the chitooligomer derivatives had low to moderate activity (MIC = 4006 to 512 µg/mL) and the chitosan derivatives had moderate to high activity (MIC = 523 to 32 µg/mL). These previous results other antimicrobial chitosan derivatives were therefore generally consistent with the current study of the M_w_ activity relationship for TMC. The antimicrobial activity of chitosan derivatives is also influenced by the chemical structure, degree of acetylation, charge etc. The influence of these is discussed in detail in a recently published review paper [13].

In contrast to the few studies on chitosan derivatives, there are a number of studies that have reported the M_w_ antimicrobial activity relationship for unmodified chitosan as discussed in the introduction. These studies have looked at different M_w_ ranges and activity against different microorganisms. These authors reached somewhat different conclusions about the M_w_ activity relationship and none of these studies have defined a specific M_w_ that is comparable to CMW in the current study. However, a closer investigation of the previously reported data shows that the findings of the previous studies are not inconsistent with the current study. Similar relationships between M_w_ and activity of chitosan have been observed but the conclusions vary partially because of differences in the focus of these studies and in the M_w_ range that has been investigated. 

Two of these studies have started from a relatively high M_w_. No et al. [9] studied activity properties of chitosan with Mw ranging from 28 to 1671 kDa and in most cases the materials had very similar activity with MIC values ranging from 0.05% to 0.1% and there was no clear trend with M_w_. However, the 28 kDa and 1106 kDa materials were found to be less active (MIC > 0.1%) against about half of the bacteria. Younes et al. [11] studied molecular weight 42 kDa to 135 kDa and DA ranging from 2% to 61%. The N-acetylation (DA) had a very significant negative effect on antimicrobial activity but the trend for molecular weight was much less marked. In most cases, there is only about a twofold difference (1 dilution) or no difference in activity between the highest and lowest M_w_ chitosan with identical DA. This is a small difference relative to the current study where there was a 1000-fold difference in activity and between the lowest M_w_ and the highest M_w_ material in the case of TMC and an 8-fold difference in case of chitosan. Li et al. [8] on the other hand reported only on the activity of low molecular weight COS with M_w_ 2 kDa and less. In their study, the M_w_ had a very significant effect on activity. There was a very clear trend where the activity increased with M_w_. The difference in activity between the most active and highest M_w_ material (>2 kDa) and the lowest molecular weight material (0.82 kDa), which showed activity, was 64 fold. Although Mellgård et al. [10] could not observe any trend in antibacterial activity with M_w_, they did report that 2KDa chitosan was inactive in all cases but one where it showed low activity against *E. coli.* Chitosan materials with a higher molecular weight (M_w_ 28 to 224 M_w_) were, on the other hand, active against the tested bacteria in most cases. Omura et al. [12] also found that COS with M_w_ 0.18 to 0.99 kDa were inactive, whereas chitosan samples with M_w_ 11 to 400 kDa were active and they could not report any clear trend in activity for the higher M_w_ material. Kulikov et al. [6] reported that COS with M_w_ 0.7 and 1.5 kDa are inactive against all *Candida* species except one and that chitosan with M_w_ 8.4 to 70 kDa is more than 10 times more active than the low molecular weight COS, except for *C. albicans* where the difference was less but still more than fourfold. The activity was in most cases the same for 6, 8.4, 9.7, 20.0 and 70 kDa chitosan and in this M_w_ range, there was no trend in the activity. 

The general findings of these previous studies of chitosan are therefore very similar to the current study of TMC and chitosan. Thus, these previous studies also indicate that there is initially a very marked increase in activity with M_w_ below a certain molecular weight and above this value the M_w_ has limited or no effect on activity. This molecular weight value is therefore comparable to CMW as we have defined it in the current study. These previous studies also suggest that CMW for antimicrobial activity may, in general, be less than the value (50 kDa) found for the activity of chitosan against *S. aureus* in the current study and more similar to the CMW value we found for TMC.

## 4. Materials and Methods

### 4.1. Materials

Chitosan TM 4138 (M_w_ = 225 kDa, and DA = 8%) was obtained from Primex ehf (Siglufjördur, Iceland). The degree of acetylation (DA) was obtained from the integrals of the ^1^H NMR spectrum and the weight average molecular weight (M_w_) was calculated using size exclusion chromatography. All chemicals were purchased from Sigma–Aldrich and used without any further purification. Dialysis membrane (RC, Spectra/Por, *M*_w_ cutoff 3500 Da) was purchased from Spectrum^®^ Laboratories Inc. (Rancho Dominguez, CA, USA). *Staphylococcus aureus* (ATCC 29213) *was* obtained from the American Type Culture Collection. Mueller–Hinton Broth and blood agar [heart infusion agar with 5% (*v*/*v*) defibrinated horse blood] were purchased from Oxoid (Basingstoke, Hampshire, UK).

### 4.2. Synthesis

#### *N,N,N*-Trimethyl Chitosan

Chitosan (4 g, 24.4 mmol) was stirred in DMF-H_2_O (100 mL, 1:1), followed by addition of NaOH (2.9 g, 73.4 mmol, 3 eq) under cooling (10 °C). Methyl iodide (9.15 mL, 146.9 mmol, 6 eq) was then added dropwise to the reaction mixture. After completion of addition, the reaction mixture was allowed to reach room temperature and then stirred for 48 h. The reaction mixture was then precipitated with acetone (150 mL) and the solid obtained was collected by filtration, followed by washing with fresh acetone (2 × 150 mL). The product was then subjected to the same reaction conditions three more times. The final product obtained was first purified by ion-exchange with 10% NaCl solution followed by dialysis for 2 days and freeze-drying. The pure product was obtained as a white solid. Yield: 4.7 g (80.0%). 

FT-IR (KBr): 3239 (O–H), 2927 (C–H), 1636 (C=O amide I), 1473 (C–H), 1030 (C–O–C), 880 (C–H) cm^−1^. ^1^H NMR (400 MHz, D_2_O): δ 2.01 (NAc), 3.28 [N(CH_3_)_3_], 3.68–4.07 (H-2, H-5, H-6), 4.30–4.42 (H-3, H-4), 5.42 ppm (H-1).

### 4.3. Hydrolysis

#### Hydrolysis of TMC and Chitosan

TMC (1g, 4.92 mmol) was stirred in (a) concentrated HCl solution (7.5 mL) at 30 °C and samples (0.5 mL) were collected at time intervals of 0.15 h, 0.5 h, 1 h, 2 h, 4 h, 8 h, 24 h, 48 h, 72 h, 120 h, 216 h, 384 h, and 528 h and (b) in 1 M HCl solution (7.5 mL) at 60 °C and samples (0.5 mL) were collected at time intervals of 0.15 h, 0.5 h, 1 h, 2 h, 4 h, 8 h, 24 h, 48 h, 72 h, 120 h, and 216 h. Chitosan (1 g, 6.25 mmol) was stirred in (a) concentrated HCl (7.5 mL) at 30 °C and samples (0.5 mL) were collected at time intervals of 0.15 h, 0.5 h, 1 h, 2 h, 4 h, 8 h, 24 h, 48 h, 72 h, 120 h, 216 h, 384 h, 528 h, 884 h, and 1388 h and (b) in 1 M HCl solution (7.5 mL) at 60 °C and samples (0.5 mL) were collected at time intervals of 0.15 h, 0.5 h, 1 h, 2 h, 4 h, 8 h, 24 h, 48 h, 72 h, 120 h, and 216 h. The samples were precipitated in acetone (60 mL), filtered and dried in vacuum for 3 h.

### 4.4. Characterization

Characterization of the compounds was performed using ^1^H-NMR, COSY-NMR and IR spectroscopy. ^1^H-NMR spectra were recorded using a Bruker Avance 400 instrument operating at 400.13 MHz at 300 K. NMR samples were prepared in D_2_O at concentrations of 10–15 mg/mL. The *N*-acetyl peak (2.08 ppm) was used as the internal reference in all spectra. FT-IR measurements were performed with a Nicolet iZ10 FT-IR instrument (Thermo Scientific Corporation, Bartlesville, OK, USA). Samples were placed over a diamond crystal and the spectrum was recorded using OMNIC 7 software. Each spectrum was collected in transmittance using 32 scans, resolution of 4 and data spacing of 0.482 cm^−1^. 

Equivalent quantities of reagents were calculated using one glucosamine unit. The degree of substitution (DS) for the *N*,*N*,*N*-trimethyl chitosan derivative was evaluated on the basis of the integral values in ^1^H NMR. The following equations were used for calculating the degree of acetylation (DA) and the degree of substitution (DS) for the chitosan derivatives:
(1)Degree of acetylation, $DA=\left[\frac{\int HAc}{\int H2-H6}\times \frac{6}{3}\right]\times 100\;\left(1\right)$(2)DS for *N,N,N*-trimethylation, $DS_{TRI}=\left[\frac{\int N-CH_{3}}{\int H2-H6}\times \frac{6}{9}\right]\times 100\;\left(2\right)$
where H2 and H6 are the protons at position 2 and 6 respectively in the glucosamine unit; HAc is the protons in the *N*-acetyl group; *N*–CH_3_ are the protons in the *N*,*N*,*N*-trimethyl group.

### 4.5. Molecular Weight Determination

GPC measurements were done using the Polymer Standards Service (PSS) (GmbH, Mainz, Germany), Dionex Ultimate 3000 HPLC system (Thermo Scientific-Dionex Softron GmbH, Germering, Germany), Dionex Ultimate 3000 HPLC pump and Dionex Ultimate 3000 autosampler (Thermo Scientific-Dionex Softron GmbH, Germering, Germany), Shodex RI-101 refractive index detector (Shodex/Showa Denko Europe GmbH, Munich, Germany) and PSS’s ETA-2010 viscometer. WINGPC Unity 7.4 software (PSS GmbH, Mainz, Germany) was used for data collection and processing. The eluent used was 0.1 M NaCl/0.1% TFA solution. Each sample was dissolved in the same eluent as mentioned above at a concentration of 1 mg/mL, filtered through a 0.45 μL filter (Spartan 13/0.45 RC, Whatman, Little Chalfont, Buckinghamshire, UK) before measurement at 25 °C using a flow rate of 1 mL/min. Each sample had an injection volume of 100 µL and a retention time of 25 min and all the measurements were done in triplicates.

### 4.6. Antibacterial Tests

The antibacterial activity was tested against the Gram-positive bacteria *S. aureus* (ATCC 29213). The broth microdilution method was used to determine the MIC values using Mueller–Hinton broth at pH 7.2 for TMC and 5.5 for chitosan. Samples for TMC were prepared in sterile water (pH = 7.2) while chitosan samples were prepared at pH = 5.5 at an initial concentration of 32,768 µg/mL. This was then serially diluted by twofold dilutions in a 96-well plate using Muller–Hinton broth, to get final concentrations varying from 16,384 to 4 µg/mL. Gentamicin was used as the performance control during the test. A standard 0.5 McFarland suspension (1–2 × 10^8^ colony-forming units (CFU)/mL) was prepared by direct colony suspension in Mueller–Hinton broth. This suspension was further diluted to achieve a final test concentration of 5 × 10^5^ CFU/mL in the wells of the microtiter plate. Mueller–Hinton broth without any addition was used as a sterility control and Mueller–Hinton broth with bacterial suspension as a growth control. The microtiter plates were then incubated at 35 °C for 18 h under moistened conditions. The MIC values were determined as the lowest concentration of the antibacterial agent that completely inhibited the visible growth of the microorganism in the microtiter plate. 

## 5. Conclusions

*N*,*N*,*N*-trimethyl chitosan (TMC) with a very high degree of *N*,*N*,*N*-trimethylation and low O-methylation was synthesized from chitosan starting material with DA of 8%. 

Acidic hydrolysis was performed on TMC and chitosan material and to produce samples of different M_w_ for studying the relationship between antibacterial activity and M_w_. The polymers were treated at different temperatures with concentrated (12 M) or diluted (1 M) HCl. Rapid hydrolysis of the polymer was observed in the first 24 h with more than 50% reduction in the M_w_. Continued incubation up to 1388 h resulted in slow further degradation. NMR analysis confirmed the hydrolysis of the glycosidic bond. *N*,*N*,*N*-Trimethylation (DS_TRI_) was not affected and no other change in the structure was observed with extensive acid treatment. 

The hydrolysis of chitosan gave 26 samples with M_w_ ranging from 133 to 2 kDa and 24 samples of TMC with M_w_ ranging from 144 to 2 kDa. 

The antimicrobial assay was performed to determine the MIC for activity against *S. Aureus* with all samples, as well as glucosamine.

The MIC values for chitosan and TMC against *S. aureus* decreased with M_w_ until a certain M_w_ value (CMW) was reached. After this, no change in activity with a further increase in M_w_ was observed. The CMW was determined to be 20 kDa for TMC. The activity of chitosan also increased with M_w_ up to a CMW value of 50 kDa after which the M_w_ had no effect on activity

This is the first study of molecular weight activity relationship for TMC. A number of studies have reported a similar pattern of activity for different M_w_ chitosan against different microorganism and the relationship for other chitosan derivatives may also be similar. 

These results on the Mw activity relationship may have implication for the possible mechanism of action for TMC and chitosan. This will be investigated in further studies.

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
