# Peer review of "The Effect of Molecular Weight on the Antibacterial Activity of *N*,*N*,*N*-Trimethyl Chitosan (TMC)"

_ijms, 2019, doi:10.3390/ijms20071743_

Round 1
Reviewer 1 Report
The authors investigated the antimicrobial properties of different samples based on chitosan and N,N,N, trimethyl chitosan, having different molecular weights (MW), in order to find a correlation between molecular weight and biological activity. Such samples, at different MW, were obtained by acidic hydrolysis carried out at different times with two different HCl concentrations.
Although interesting results were obtained for the Mw antimicrobial activity relationship (determination of a MW threshold (CMW) for chitosan and their chitosan derivatives), the authors should better stress the implications that such results have on the antimicrobial activity of their samples. They, in fact, should provide some hypothesis on the action mechanism of their polymers versus the tested microorganism.
The numeration of the figures should be checked.
All references should be reported in the journal style (see paper titles).
The English should be revised.
The following items should be addressed:
Results
Section “Structural characterization of hydrolysed products”.
The authors should better discuss the FTIR data.
For example, the peak at 2879 cm-1 is only due to CH stretching and not to vibration of NH group. While, the peak at 1508 cm-1 corresponds to the NH bending of the primary amine not protonate of chitosan. This absorption is then absent in the TMC spectrum as a result of the change of the primary amine in quaternary amine. Also, the peak at 1473 cm-1 is indicative of the quaternization reaction (methyl asymmetrical C–H bending).
Section “Mw antibacterial property relationship”
In the table 2, the MIC of gentamicin should be also reported.
Figure 5. What is the DP reported in the x2 axis?
Discussion
Section “Mw antibacterial property relationship”
The discussion of the literature data is neither clear or exhaustive. The antimicrobial activity of chitosan and chitosan derivatives depends on several parameters: acetylation degree, molecular weight, presence of quaternary groups, type and amount of quaternary groups, positive charge in backbone or in side chain of chitosan, ecc.. Therefore, the data discussion should be done in systematic way considering the relationship between each of that parameters and molecular weight.
Line 279-281- “The difference in activity….that showed activity was 8-64 fold”. Clarify the sentence.
Materials and Methods.
Section “Hydrolysis”.
The authors should bring together all the hydrolysis procedures into a single paragraph highlighting the differences for the samples (different T, HCl concentration, sampling intervals).
For example, the sampling intervals during the hydrolysis reaction could be reported in a sentence of the type “the samples were collected at time intervals from 0.15 to 1388 h depending on the type of sample”.
In this section, moreover, it should be added the amount in mmol of chitosan and TMC employed.
Conclusion.
Line 387. Eliminate “elevated” temperatures and insert “different” temperatures.
Line 396. “The MIC values for chitosan….. increased up to…..” This statement is wrong. Correct the concept.
Author Response
The authors investigated the antimicrobial properties of different samples based on chitosan and N,N,N, trimethyl chitosan, having different molecular weights (MW), in order to find a correlation between molecular weight and biological activity. Such samples, at different MW, were obtained by acidic hydrolysis carried out at different times with two different HCl concentrations.
Although interesting results were obtained for the Mw antimicrobial activity relationship (determination of a MW threshold (CMW) for chitosan and their chitosan derivatives), the authors should better stress the implications that such results have on the antimicrobial activity of their samples. They, in fact, should provide some hypothesis on the action mechanism of their polymers versus the tested microorganism.
Response. We grateful for the thoughtful comments and corrections made by the Reviewer 1. – We certainly agree that our results on the structure-activity relationship have implication for the possible mechanism of action and we have now added a comment on this in the conclusion. However, we would like to avoid suggesting a hypothesis at this stage. – We are now working on a second manuscript based on careful reanalysis of literature data for the different type of microorganisms and in that manuscript, we will discuss the implication for the mechanism of action in some detail.
The numeration of the figures should be checked.
Response: The Figure numbering has been corrected in line 171
All references should be reported in the journal style (see paper titles).
Response: We thank the reviewer for noticing this. We have corrected the styles where needed.
The English should be revised.
Response: The manuscript has been checked and revised by a native language expert with considerable experience in scientific writing. Now we have done further checking and corrections before the submission of the revision.
The following items should be addressed:
Results
Section “Structural characterization of hydrolysed products”.
The authors should better discuss the FTIR data.
For example, the peak at 2879 cm-1 is only due to CH stretching and not to vibration of NH group. While, the peak at 1508 cm-1 corresponds to the NH bending of the primary amine not protonate of chitosan. This absorption is then absent in the TMC spectrum as a result of the change of the primary amine in quaternary amine. Also, the peak at 1473 cm-1 is indicative of the quaternization reaction (methyl asymmetrical C–H bending).
Response: We have corrected and revised the discussion (Line 187-195) of FTIR data as suggested by the reviewer
Section “Mw antibacterial property relationship”
Response: We have included the MIC for gentamicin in Table 2.
In the table 2, the MIC of gentamicin should be also reported.
Figure 5. What is the DP reported in the x2 axis?
Response: DP = degree of polymerization. This explanation has been included in the caption for Figure 5.
Discussion
Section “Mw antibacterial property relationship”
The discussion of the literature data is neither clear or exhaustive. The antimicrobial activity of chitosan and chitosan derivatives depends on several parameters: acetylation degree, molecular weight, presence of quaternary groups, type and amount of quaternary groups, positive charge in backbone or in side chain of chitosan, ecc.. Therefore, the data discussion should be done in systematic way considering the relationship between each of that parameters and molecular weight.
Response: We have recently published a review paper where we include an extensive discussion of the influence of all these parameters on antimicrobial activity (See:). In the present manuscript, we have focused only on the influence of Mw on activity and therefore we have avoided detailed discussion of other parameters. – We have now added a sentence noting that the influence of other parameters is discussed in the review paper.
Line 279-281- “The difference in activity….that showed activity was 8-64 fold”. Clarify the sentence.
Response: We thank the reviewer for pointing out an unclear sentence. This sentence has been corrected.
Materials and Methods.
Section “Hydrolysis”.
The authors should bring together all the hydrolysis procedures into a single paragraph highlighting the differences for the samples (different T, HCl concentration, sampling intervals).
For example, the sampling intervals during the hydrolysis reaction could be reported in a sentence of the type “the samples were collected at time intervals from 0.15 to 1388 h depending on the type of sample”.
In this section, moreover, it should be added the amount in mmol of chitosan and TMC employed.
Response: We have combined all the hydrolysis procedures into one paragraph and reported the sample collection time intervals in one sentence. We have also added the mmol for chitosan and TMC.
Conclusion.
Line 387. Eliminate “elevated” temperatures and insert “different” temperatures.
Response: This has been done.
Line 396. “The MIC values for chitosan….. increased up to…..” This statement is wrong. Correct the concept.
Response: We thank the reviewer for pointing out the unfortunate error. This sentence has now been corrected.
Reviewer 2 Report
1. Authors need to provide the agar plate images for different Mw of TMC and Chitosan.
2. Figure 2-5 captions are need to re-write.
3. Need to explain CFU unit properly, where it has used in first time.
4. Page 10, line 294, Authors used abbreviation of CMW, but it has not suitable properly.
5. Figure 3, inside texts are not visible, should be improve.
6. There is not lot of grammatical errors, should be correct carefully.
7. In the materials section, authors included the chemical company details without removal of web link, should be take care these things importantly.
8. The conclusion part is need to be rewrite.
Author Response
Comments and Suggestions for Authors
We thank the reviewer 2 for good comments and careful review.
1. Authors need to provide the agar plate images for different Mw of TMC and Chitosan.
Response: In this work, we only measured the MIC and thus there are no agar plate images to show. This was done to save time with a relatively large number of samples. – Our previous work (fx: Rúnarsson, Ö. V.; Holappa, J.; Malainer, C.; Steinsson, H.; Hjálmarsdóttir, M.; Nevalainen, T.; Másson, M., Antibacterial Activity of N-Quaternary Chitosan Derivatives: Synthesis, Characterization, and Structure-Activity Relationship Investigations (SAR). Eur. Polym. J. 2010, 46 1251-1267 and Sahariah, P.; Gaware, V. S.; Lieder, R.; Jónsdóttir, S.; Hjálmarsdóttir, M. Á.; Sigurjonsson, O. E.; Másson, M., The Effect of Substituent, Degree of Acetylation and Positioning of the Cationic Charge on the Antibacterial Activity of Quaternary Chitosan Derivatives. Marine Drugs 2014, 12, (8), 4635-4658.) has shown that the MLC value (found be colony counting on agar plates) is very similar to the MIC with generally less than 1-2 dilutions difference,
2. Figure 2-5 captions are need to re-write.
Response: We have rewritten the captions for Figures 2-5.
3. Need to explain CFU unit properly, where it has used in first time.
Response: We have added the definition.
4. Page 10, line 294, Authors used abbreviation of CMW, but it has not suitable properly.
Response: This sentence has been changed
5. Figure 3, inside texts are not visible, should be improve.
Response: We have enlarged the texts in the Figure.
6. There is not lot of grammatical errors, should be correct carefully.
Response: The manuscript has been checked and revised by a native language expert and now we have done further checking and corrections before the submission of the revision.
7. In the materials section, authors included the chemical company details without removal of web link, should be take care these things importantly.
Response: We thank the reviewer for careful review and noticing this, which we had missed. This has been changed.
8. The conclusion part is need to be rewrite.
Response: The conclusion section has been changed as also suggested by Reviewer 1.
Round 2
Reviewer 1 Report
I have appreciated the efforts of the authors to amend the manuscript. The authors have accepted almost all of suggestions proposed by the reviewer. The quality of the manuscript have been sufficiently increased.
Reviewer 2 Report
The revised version is suitable for publication.